# High Adherence to the Mediterranean Dietary Pattern Is Inversely Associated with Systemic Inflammation in Older but Not in Younger Brazilian Adults

**DOI:** 10.3390/nu16091385

**Published:** 2024-05-02

**Authors:** Amália Almeida Bastos, Paula Victória Félix, João Valentini Neto, Marcelo Macedo Rogero, Regina Mara Fisberg, Mary Yannakoulia, Sandra Maria Lima Ribeiro

**Affiliations:** 1Department of Nutrition, Public Health School, University of Sao Paulo, Sao Paulo 01246-904, Brazil; paula.victoria@gmail.com (P.V.F.); joaoneto@usp.br (J.V.N.); mmrogero@usp.br (M.M.R.); regina.fisberg@gmail.com (R.M.F.); smlribeiro@usp.br (S.M.L.R.); 2Department of Nutrition and Dietetics, School of Health Sciences and Education, Harokopio University, 17671 Athens, Greece; myianna@hua.gr

**Keywords:** younger adults, older adults, adherence, Mediterranean dietary pattern, systemic inflammation

## Abstract

The Mediterranean dietary pattern (MPD) has shown promise in preventing low-grade systemic inflammation (LGSI). This study tested if a high adherence to the MDP by younger and older Brazilian adults is associated with lower LGSI and investigated which Mediterranean food components may contribute to these associations. We performed a secondary study on 2015 ISA-Nutrition (290 younger adults (20–59 years old) and 293 older adults (≥60 years old)), a cross-sectional population-based study in São Paulo, SP, Brazil. The adherence to the MDP was assessed using the Mediterranean Diet Score (MedDietScore), obtained from two non-consecutive 24 h dietary recalls (24HDRs). The LGSI score (from plasma CRP, TNF-α, and adiponectin) identified the inflammatory status. Linear regression models assessed the association between LGSI and the MedDietScore. In older adults only, a high adherence to the MDP signified an 11.5% lower LGSI score. Older adults, classified with high adherence to the MDP, differed by consuming lower meat intake and full-fat dairy. Between older adults, the intake of vegetables and olive oil was inversely associated with the levels of LGSI. Thus, among older adults, the intake of some specific Mediterranean food determined high adherence to the MDP and was associated with decreased LGSI.

## 1. Introduction

Chronic low-grade systemic inflammation (LGSI) is widely acknowledged as a potential factor contributing to chronic diseases [1]. Inadequate dietary intake, particularly in terms of nutrient excess or overnutrition, is associated with metabolic inflammation (metaflammation), which has been identified as a contributor to the predisposition for unhealthy aging [2]. In light of this, it is crucial to explore modifiable factors, selecting age groups within the dietary intake profile as part of the nutritional assessment process to determine nutritional status [3].

Given its recognized anti-inflammatory role, the MDP has been explored in non-Mediterranean populations, including Western countries. In a recent systematic review and meta-analysis [4], community-dwelling older adults with higher adherence to the MDP presented lower values of the C-reactive protein (CRP) and interleukin 6 (IL-6). Four of the five studies included in the meta-analysis were conducted in non-Mediterranean populations, including two from the United States.

Studying this topic in Western populations becomes even more relevant considering the pro-inflammatory Western-style diet due to the frequent consumption of highly processed foods [5]. The Western diet (WD) is typically rich in saturated fatty acids, sugar, salt, and food additives, acting oppositely to the anti-inflammatory effects of the Mediterranean diet [5,6]. Among Brazilians, ultra-processed foods represent, on average, 19.7% of the total calories consumed [7], while in Mediterranean countries such as Greece and Italy, the corresponding consumption is lower, at 13.7% and 13.4%, respectively [8]. Also, the common practice of using herbs and spices in the Mediterranean regions gives flavor, improves the palatability of the dishes, and increases salty taste sensitivity, allowing a reduction in salt intake [9].

A second important point when evaluating the benefits of high adherence to the MDP in a Western population is the low consumption of some foods that are abundant in Mediterranean areas. For instance, energy intake from fish represents 5.7% (mean = 13.1 g/day) of the total energy consumption among Brazilians, and the frequency intake of wine is 0.5% (mean = 1.9 g/day), whereas, in the Greek population, the mean (±SD) intake of fish by men and women was 26.4 g/day (±20.3) and 21.7 g/day (±15.4), respectively [10]. For wine, men and women who highly adhere to the MDP had an intake of 155 mL/day (±35) and 75 mL/day (±45), respectively [11].

These WD characteristics favor the lower consumption of polyphenols, monounsaturated and polyunsaturated fatty acids, and high cholesterol intake, contributing to a close connection with metaflammation [5]. In addition, regional characteristics, cultural differences, and culinary practices can compromise the benefits of the MDP on health outcomes in non-Mediterranean populations. For example, the amount of phytosterols, tocopherols, and phenolic compounds can vary between types of olive oil and the way it is used to cook; the variety and cultural preferences of fruits and vegetables can also implicate the different quantities of nutrient characteristics of the MDP; despite not being evaluated in a dietary index, drinking wine with a meal brings extra health benefits, a practice that may not be common among non-Mediterranean populations [12].

Recent studies have drawn attention to trends of lower degrees of adherence to the MDP among younger populations and reinforce how encouraging Mediterranean dietary practices positively impacts their health condition and physical and mental performance [13,14,15]. Although the most promising and cohesive results showing the benefits of the Mediterranean on health status come from longitudinal studies including Mediterranean populations, some cross-sectional and even longitudinal studies carried out in non-Mediterranean populations also shared positive findings [4,16,17]. Studies with Brazilian younger and older populations involving the Mediterranean diet and LGSI have not yet been studied using systematic reviews and meta-analysis.

Further investigation is needed to explore the characteristics of Mediterranean food intake that provide the anti-inflammatory potential of high adherence to the MDP among Brazilian samples. Additionally, it is essential to test the hypothesis that Brazilian younger adults may have a different profile of adherence to this dietary pattern than older adults, which can result in specific systemic inflammatory responses in each group. The present study aimed (i) to characterize and compare the dietary consumption concerning the degrees of adherence to the MDP in younger and older adults; (ii) to investigate if a high adherence to this dietary pattern is associated with lower levels of systemic inflammation in younger and older adults living in São Paulo, SP, Brazil; (iii) and investigate the association of the individual Mediterranean components with systemic inflammation.

## 2. Materials and Methods

### 2.1. Study Design and Participants

This study includes data from the 2015 Health Survey of São Paulo with a focus on nutrition (2015 ISA-Nutrition), a cross-sectional population-based study. People residing in permanent private households in the urban area of São Paulo city were randomly selected from September 2014 to December 2015.

Details about the 2015 Health Survey of São Paulo (ISA-Capital) sampling process can be found elsewhere [18]. In brief, the sample was stratified by clusters and performed in the following two stages: census tracts, which were stratified into five Regional Health Coordinations of São Paulo City (north, mid-west, southeast, south, and east), and households. All individuals from the families who belonged to the demographic domain selected in the study and who met the inclusion criteria (people of both genders, aged 12+ years, and residing in the urban area of São Paulo during the period of the survey) were interviewed. The exclusion criteria were individuals with chronic alcoholism and those on an enteral and/or parenteral diet.

In order to minimize the effects of losses and refusals, larger independent random selections were performed, securing a sample that allowed the estimation of proportions of the differences of 0.50. The sampling error was seven percentage points, considering a 95% confidence level and a delineation effect of 1.5.

Among the 4059 participants of the 2015 ISA-Capital, 1737 were randomly selected to participate in the first phase of 2015 ISA-Nutrition, answering the first 24 h dietary recall (24HR). In the second phase of 2015 ISA-Nutrition, 901 were invited and agreed to have their blood sample collected, underwent anthropometry, and answered the second 24HR. For the present study, data from 290 adults and 293 older adults (mean age 50.3 ± 0.79) collected in all phases of 2015 ISA-Nutrition were analyzed. More detailed information about 2015 ISA-Nutrition can be found elsewhere [19].

This study was conducted according to the guidelines laid down in the Declaration of Helsinki. The Research Ethics Committee of the School of Public Health of the University of São Paulo approved the 2015 ISA-Capital and 2015 ISA-Nutrition surveys (protocols 32344014.3.3001.0086 and 30848914.7.0000.5421, respectively. All individuals included in the survey had their data collected only after signing the Informed Consent Form.

### 2.2. Blood Collection and Investigation of the Low-Grade Systemic Inflammation

Blood samples were collected after 12 h of fasting in EDTA (Ethylenediamine tetra-acetic acid)-containing tubes and stored for posterior analyses. The plasma circulating C-reactive protein (CRP) was measured by kinetic turbidimetry using the IMMAGE^®^ immunochemistry system kit (Beckman Coulter Inc., Brea, CA, USA). The tumor necrosis factor-alpha (TNF-α) and adiponectin were measured via a multiplex immunoassay (Milliplex, Merk Millipore, Darmstadt, Germany).

Considering the different ranges of inflammatory biomarkers, to investigate the systemic inflammatory response, we created an LGSI score based on previous inflammatory scores tested in Brazilian samples [20,21]. First, the normalized inflammatory biomarker values were obtained using the Min-Max scaling technique. Then, the LGSI score was calculated by summing up the normalized pro-inflammatory biomarker values (CRP and TNF-α) and subtracting them from the sum of normalized anti-inflammatory biomarker values (adiponectin). These two steps are detailed below:(1)$$Normalizedbiomarker\left(n\_b\right)=\frac{\left\{(b-\min\left(b\right)\right\}}{\left\{\max\left(b\right)-min(b)\right\}}$$
(2)LGSIscore=(n_CRP+n_TNFα)−(n_adiponectin)

### 2.3. Dietary Assessment

Food consumption data were obtained by conducting two non-consecutive 24 h dietary recalls (24HR) within one year, covering all days of the week and seasons. The first 24HR was collected during an initial domicile visit. It was completed based on the Multiple Pass Methods (MPMs) developed by the US Department of Agriculture (USDA) [22] to reduce errors in dietary measurements [23]. The second 24HR was collected through telephone calls using the interview system incorporated into the Nutrition Data System for Research (NDSR, version 2014, developed by the Nutrition Coordination Center, University of Minnesota, Minneapolis, MN, USA) [24], which was then entered into the NDSR software (version 2014). Subsequently, habitual dietary intake was estimated using the Multiple Source Method (MSM) [25]. This statistical method estimates the usual intake of food and nutrients consumed by the population reported in the two 24HR.

### 2.4. Adherence to the Mediterranean Dietary Pattern

The adherence to the MDP was evaluated using the Mediterranean Diet Score (MedDietScore), which has a theoretical range of 0–55 points, with higher values indicating higher adherence [11]. The MedDietScore is based on the consumption of 11 main dietary components of the Mediterranean diet (non-refined cereals, fruits, vegetables, potatoes, legumes, olive oil, fish, red meat, poultry, full-fat dairy products, and wine). An individual score, from 0 to 5 points, was calculated for the 11 food components according to their recommended intake based on the traditional Mediterranean diet pyramid. For the typical food components of the Mediterranean diet (non-refined cereals, fruits, vegetables, potatoes, legumes, olive oil, and fish), a score of “zero” was given for no consumption, a score of “one” was given when the consumption ranges from 1 to 4 servings/month, a score of “two” was given for 5 to 8 servings/month, a score of “three” was given for 9 to 12 servings/month, a score of “four” was given for 13 to 18 servings/month and a score of “five” was given for more than 18 servings/month. The inverse score was applied for the consumption of foods that differed from this dietary pattern (meat and its derivatives, poultry, and full-fat dairy products). Especially for wine, points ranging from 0 to 5 were assigned according to the amount of consumption in ml per day. A score of “five” was assigned to consumption < 300 mL/day, a score of “zero” to no consumption or consumption > 700 mL/day, and a score of 4 to 1 for the consumption of 300 to 400 mL/day, 400 to 500 mL/day, 500 to 600 mL/day, 600 to 700 mL/day, respectively (assuming 100 mL at a 12 g ethanol concentration).

### 2.5. Socio-Demographic Characteristics

The sociodemographic characteristics were obtained through the application of a structured questionnaire and for the present study included the following: sex (male or female), age (20–59 or ≥60 years old), years of formal education (≤9, 10–12 or >12), per capita household income (≤1 minimum wage and >1 minimum wage) and race (white or nonwhite).

### 2.6. Behavior and Lifestyle Variables

The physical activity level was determined using the full version of the International Physical Activity Questionnaire (IPAQ), previously validated in Brazil. This variable was categorized as follows: do not comply with recommendation (<150 min/week) and comply with recommendation (≥150 min/week). These recommendations are under the World Health Organization—WHO [26]. The variable smoking status was self-reported and classified as never smoked, former smoker, or current smoker. The total energy intake (kcal) was calculated by converting the gram intake of carbohydrates, protein, and fat to calories as follows: 1 g carbohydrates = 4 kcal, 1 g protein = 4 kcal, and 1 g fat = 9 kcal.

### 2.7. Health Conditions

Body mass index (BMI) was calculated by dividing the measured weight (kg) by the squared measured height (m^2^). For adults, the classification of the BMI followed the World Health Organization’s (2000) reference as follows [27]: below normality range: <18.5 kg/m^2^; normality range: 18.5 to 24.9 kg/m^2^; overweight: ≥25 kg/m^2^ and ≤30 kg/m^2^; obese: >30 kg/m^2^). Older adults’ (>60 years old) BMI were classified according to the Pan American Health Organization [28] (below normality range: ≤23 kg/m^2^; normality range: >23 and <28 kg/m^2^; overweight: ≥28 and <30 kg/m^2^; obese: ≥30 kg/m^2^). We also investigated multimorbidity (the coexistence of two or more of the following chronic conditions [29]: diabetes mellitus, high blood pressure, hypercholesterolemia, arthritis or arthrosis or rheumatism, cardiovascular diseases, and cerebrovascular accident) and medication use (the self-reported use of medication in the last two days of the interview to treat chronic diseases).

### 2.8. Statistical Analysis

Normality was examined using the Shapiro–Wilk test and histogram analysis. Categorical variables were described by the percentage of absolute frequency; continuous variables normally distributed were presented as the mean and standard deviation (±SD), while those non-normally distributed were descriptive in the median with interquartile range (IQR). The adherence to the MDP was defined in three degrees based on percentiles of the MedDietScore to determine low adherence (≤25th percentile), moderate adherence (>25th percentile and <75th percentile), and high adherence (≥75th percentile). To investigate the association between low-grade systemic inflammation and adherence to the MDP, linear regression was fitted in four different models (crude and adjusted models). The adjusted models were defined following the hierarchical model for analyzing factors associated with low-grade systemic inflammation. Model 1 was adjusted for sociodemographic confounders (sex, age, race, and education) that compose distal factors in the hierarchical model, while Models 2 and 3 were adjusted for intermediate (lifestyle variables—physical activity and smoke status) and proximal factors (health condition—BMI, multimorbidity and medication use) (Appendix A).

For all the analyses, the statistical significance was investigated, considering a *p*-value < 0.05. The analyses were undertaken in Stata software version 14.0 (StataCorp, College Station, TX, USA).

## 3. Results

### 3.1. Sociodemographic, Lifestyle and Health Characteristics

Among the 290 younger adults and 293 older adults included in this study, 54.2% and 52.2% were male, had a median age of 41 (33–50) and 68 (63–73) years old, were predominantly white (52.4% and 61.6%) and reported a household income greater than 1 minimum wage (51.3% and 58.9%), respectively. Younger adults with between 10 and 12 years of formal education (36.9%) were more prevalent, while 64.4% of older adults referred to ≤9 years of formal education. Most of both groups did not meet the recommendations for physical activity (≥150 min/week) and never smoked. Regarding health conditions, 36.7% of younger adults and 38.1% of older adults were within the BMI normality range, and 83.8% of younger adults had no multimorbidity vs. 50.4% of older adults living with this condition. While 53.9% of younger adults did not use anti-inflammatory medication, 84.1% of older participants used at least one (Table 1).

### 3.2. Association between Adherence to the Mediterranean Diet and Low-Grade Systemic Inflammation

Regarding the adherence to MDP by the whole sample, the mean (±SD) ranged from 25 (±1.71) to 34 (±1.84) points between low and high degrees of adherence to the MDP based on the MedDietScore. As described in Table 2, four extra points between low and moderate adherence to the MDP were insufficient to maintain lower inflammatory levels in any linear regression models. On the other hand, with nine points extra, a high adherence, compared to low adherence, was directly and significantly associated with 9.5% low-grade systemic inflammation (β = −0.095; *p* = 0.008) even after full adjustment (Model 3).

When selecting the sample by age group, moderate adherence predominated among younger (40%) and older adults (38%), and the mean (±SD) of MedDietScore, representing a high adherence to the MDP, were similar for both groups (34 ± 1.75 and 34 ± 1.82). Only between older adults did high adherence, compared to low adherence, appear protective against LGSI (Table 3). There was no significant association in any linear regression model analyzing adult data. For older adults, in the fully adjusted linear regression model (Model 3), we observed an 11.5% lower LGSI level attributed to a high adherence to the MDP (β = −0.115; *p* = 0.040).

### 3.3. MedDietScore’s Food Component Intake According to the Degree of Adherence by Age Group and Its Association with Systemic Inflammation

Table 4 demonstrates the differences in the consumption of Mediterranean food components between the degrees of adherence by age group. Younger adult participants with high adherence to the MDP consumed a greater amount of non-refined cereals with potatoes, vegetables, fruits, fish, olive oil, and wine; and had a lower intake of poultry. These same differences were found when comparing older adults with low and high adherence to MDP, except for wine, the intake of which did not differ between the degrees of adherence and the lower intake of meat and full-fat dairy.

None of MedDietScore’s foods were relevant to contribute to a lower level of LGSI among younger adults after the adjustment for confounders. For older adults, the intake of vegetables and olive oil corresponding to high adherence to the MDP was individually associated with lower levels of systemic inflammation. Consuming more than ≥12 servings/week of vegetables, compared to an intake of ≤9 servings/week, is associated with a 13.2% lower LGSI score (β = −0.132; *p* = 0.010). Additionally, for older adults, an intake of ≥3 servings/week of olive oil compared to an intake of ≤1 serving/week was shown to keep lower levels of systemic inflammation by 11.5% (β = −0.115; *p* = 0.034) (Table 5).

## 4. Discussion

This cross-sectional population-based study investigated adherence to the MDP and inflammatory status data from Brazilian younger and older adult samples. Even with the existing environment and regional and cultural differences that can limit the adherence to the MDP and its health benefits in non-Mediterranean populations, our data showed that high adherence to the MDP in Brazilian older adults was able to attenuate systemic inflammation. Our results showed differences in the profile of high adherence to the MDP between younger adults and older adults, which were favorable to protection against LGSI only among older participants. Besides a greater intake of non-refined cereals with potatoes, vegetables, fruits, fish, and olive oil and a lower poultry intake observed among younger adults with high adherence, older adults with the same degree of adherence had a lower intake of meat and full-fat dairy.

Few Brazilian studies using robust samples show this positive evidence around the possible protective role of the Mediterranean diet on systemic inflammation. In a previous study from our group, using cross-sectional data (baseline) from 73 community-dwelling older adults living in the city of São Paulo, Brazil, who participated in a randomized clinical trial, the performance of dietary indexes was compared regarding their anti-inflammatory potential [21]. An anti-inflammatory index (AII) from the IL-10/IL-6 ratio was calculated; in contrast to the dietary inflammatory index (DII) and the ratio between unprocessed or minimally processed/ultra-processed food (UPR), the Mediterranean Diet Scale (MDS) was positively and significantly associated with 24.7% AII higher values (β = 0.247; 95%CI = 0.005 to 0.163). Contrary to our study, the score of adherence to the MDP obtained via the MDS was analyzed only as a continuous variable; therefore, participants were not classified by degrees of adherence. However, the median (4.0; 1.0–7.0) obtained in the MDS indicated participants’ moderate adherence to this dietary pattern. Unexpectedly, the UPR did not modify this finding. Nonetheless, the authors reinforce the relevance and need for studies on this topic in more robust samples.

Another study was conducted in molecularly proven familial hypercholesterolemia adults from Brazil (n = 92; mean age of 45 years) and Spain (n = 98; mean age of 46.8 years) to verify the association between adherence to the Mediterranean diet and biomarkers of dyslipidemia and low-grade inflammation. Considering the whole sample, strong adherence to the Mediterranean diet was significantly and inversely associated with high sensitivity CRP. This analysis was not stratified by country, besides the relevant differences observed in dietary patterns between individuals from both countries, since most Brazilians had low adherence (83.7%), while most participants from Spain were classified with strong adherence (37.8%), and HS-CRP Brazilian concentrations were higher (1.6 mg/L vs. 0.8 mg/L; *p* < 0.001) [31].

Regarding the details of adherence to the MDP by younger adults and older adults, our findings followed the results from a study using data from 4011 adults over 18 years old collected by the 2013–2014 Greek National Health and Nutrition Survey (HYDRIA). Similarly, older adults had a better profile of adherence since 39.7% and 19.5% of the participants over 65 years old had high and low adherence to the MDP, respectively, while only 25.5% of those under 65 years old presented high adherence and 35.8% had low adherence to the MDP. In general, without considering the degrees of adherence, the usual daily intake of vegetables, fruits, legumes, dairy products, fish, olive oil, and wine was higher between older men and women. Younger adults aged from 18 to 64 years old had a greater intake of red meat, cereals, alcoholic beverages different from wine, and sugar products. These characteristics contributed to a higher intake of fiber and MUFA and a lower intake of SFA (percentage of the total energy intake) by older adults [32]. In another study involving 1993 community-dwelling older people (≥65 years old) from two cities in Greece, Athens and the city of Larissa, the results showed that high adherence to the MDP by the younger and elderly brought benefits relating to their social environment which was associated with social contacts with more interactions and intellectual and physical activities [33].

Although it has not been investigated, in addition to this difference in the profile of adherence to the MDP between younger adults and older adults, the consumption of typically Western foods may have also contributed to the favorable results found only among older adults. For instance, 19.5% of the total energy intake in Brazilians aged between 18 and 59 years is represented by their UPF intake; when it comes to older adults (≥60 years old), this consumption reduces to 15.1% [7].

Following the characteristics of high adherence to the MDP by older adults, which differed from younger adults for the lower consumption of red meat and full-fat dairy, it indeed contributed to better nutritional adequacy and anti-inflammatory potential. High adherence to the MDP is commonly associated with lower and healthier fat contributing to energy intake, prioritizing MUFA over SFA, in addition to differentiating itself concerning the consumption of bioactive compounds and the antioxidant capacity of the diet [34,35]. The anti-inflammatory benefit resulting from high adherence by older adults has the protective role of greater consumption of olive oil and vegetables. Short-chain fatty acids (SCFA) derived from the bacterial fermentation of dietary fibers found in vegetables promote a reduction in intestinal permeability and pH, inhibiting the growth of Gram-negative bacteria that produce lipopolysaccharide (LPS) as an activator of systemic inflammation [36]. Also found in vegetables, carotenoids, vitamins, and minerals are recognized for their ability to inhibit leukocyte adhesion molecules and prevent nuclear factor kappa B (NF-κB) activation [37,38]. These anti-inflammatory functions are also played by oleic acid, the major lipidic component of olive oil [39,40].

Even observing a consumption profile far from that observed in the high adherence by Mediterranean populations, older people also showed benefits on their inflammatory profile. For instance, red meat consumption was limited to 1 serving per week; olive oil consumption was approximately 7 and 8 servings per week by men (n = 1514) and women (n = 1528) from Greece [11] and fish consumption by the Greek elderly was 0.7 servings per day [33]. Furthermore, even MedDietScore’s food components that differed in quantity intake between low and high adherence did not prove to be individually protective against LGSI. These findings reinforce that the benefit found among older adults comes from the interaction between all food components, which represents and makes the study of dietary patterns more relevant [41].

Maintaining good health conditions is more challenging for older adults. Independent or not by health conditions, aging may be accompanied by relevant physiological changes like the loss and impairment of teeth, a decrease in the secretion of saliva, and weakening smell and taste [42]. These conditions affect eating habits and contribute to dietary deficiencies, especially energy and protein intake [43]. Systemic inflammation can intensify this process since, in addition to their anorectic effect, the cytokines compromise protein availability, contributing to malnutrition [44,45]. The serum inflammatory markers are controlled by greater adherence to the MDP, helping to prevent, for instance, loss of muscle mass, osteoporosis, sarcopenia, frailty, and cognitive decline [46,47,48]. In addition, outside the community-dwelling environment, lower adherence to the Mediterranean diet in elderly patients when hospitalized is associated with a longer length of stay and higher circulating interleukin-6 and tumor necrosis factor alpha [49].

Using the dietary pattern approach to propose dietary recommendations to manage the health conditions of older individuals is a superior strategy that is well supported by studies. Although assessing representative dietary habits of the Mediterranean diet is regularly cited and confirmed as an effective support for healthy aging [43,50], it is very challenging for people from other geographical regions to incorporate these dietary characteristics due to cultural and socioeconomic aspects and social behaviors [51]. The results presented by studies that show this difference in the profile of adherence to the MDP by different age groups are relevant to support this process.

## 5. Limitations

This study has some limitations and several strengths. Like all cross-sectional studies, it is impossible to define causality and to generalize these results for other regions of Brazil. The 2015 ISA-Nutrition study is a relevant epidemiological investigation conducted in a representative sample of the most significant center in Brazil. Access to all data followed previously defined protocols, and food intake was assessed by two 24HR. The usual consumption of the population was used to calculate adherence to the MDP. As far as we know, this is the first study comparing and characterizing adherence to MDP among younger and Brazilian older adults in the context of systemic inflammation.

## 6. Conclusions and Future Perspectives

In conclusion, our results showed a better profile of high adherence to the MDP by older adults, differing from younger adults in terms of the lower consumption of red meat and full-fat dairy. These characteristics provided positive results of high adherence to this dietary pattern against systemic inflammation. Only among older adults was the higher consumption of vegetables and olive oil individually associated with lower levels of inflammation. Although the sample’s consumption characteristics of high adherence do not represent the traditional Mediterranean diet in quantitative terms, this study brings a reflection, especially for future Brazilian research around possible Mediterranean diet recommendations to encourage practices of this dietary pattern closer to their current and real dietary habits.

## Figures and Tables

**Table 1 nutrients-16-01385-t001:** Characteristics of the study sample according to age group. São Paulo, Brazil, 2015.

	Younger Adults (n = 290)	Older Adults (n = 293)
Variables	n (%) or Median (IQR)	n (%) or Median (IQR)
Sex		
Male	150 (54.2)	146 (52.2)
Female	140 (45.8)	147 (47.8)
Age (Years)	41 (33–50)	68 (63–73)
Race		
White	148 (52.4)	168 (61.6)
Nonwhite	142 (47.6)	119 (38.4)
Years of Education		
≤9	104 (33.9)	203 (64.4)
10–12	111 (36.9)	36 (13.3)
>12	75 (29.2)	51 (22.3)
Household income ^a^		
≤1 MW	134 (48.7)	110 (41.1)
>1 MW	123 (51.3)	141 (58.9)
Physical activity		
<150 min/week	229 (77.3)	241 (83.2)
≥150 min/week	61 (22.7)	50 (16.8)
Smoking status		
Never smoke.	172 (60.8)	175 (62.1)
Former smoker	57 (19)	80 (25.1)
Current smoker	61 (20.2)	35 (12.8)
Energy intake (kcal)	1804.5 (1516.8–2221.4)	1565.4 (1311.5–1853.3)
BMI (kg/m^2^) ^b^		
Below normality range	9 (3.6)	55 (16.7)
Within normality range	106 (36.7)	107 (38.1)
Overweight range	102 (35.2)	42 (15.3)
Obesity range	70 (24.5)	85 (29.9)
Multimorbidity ^c^		
No	239 (83.8)	144 (49.6)
Yes	50 (16.2)	144 (50.4)
Anti-inflammatory medication use ^d^		
No	151 (53.9)	48 (15.9)
Yes	133 (46.1)	242 (84.1)

^a^ MW: minimum wage value in 2015 was 224.02 USD; ^b^ BMI: body mass index (Adults: below normality range: <18.5 kg/m^2^; Normality range: 18.5 to 24.9 kg/m^2^; Excess body weight: ≥25 kg/m^2^; Older adults: below normality range: <23 kg/m^2^; Normality range: 23 to 27.9 kg/m^2^; Excess body weight: ≥28 kg/m^2^); ^c^ The coexistence of two or more chronic conditions: diabetes mellitus, high blood pressure, hypercholesterolemia, arthritis, arthrosis or rheumatism, cardiovascular diseases, and cerebrovascular accident; ^d^ Self-reported use of medication in the last two days of the interview to treat chronic diseases.

**Table 2 nutrients-16-01385-t002:** Linear regression results between low-grade systemic inflammation and MedDietScore categorized by degrees of adherence. São Paulo, Brazil, 2015.

	Degrees of Adherence (MedDietScore)				
	Low Adherence	Moderate Adherence		High Adherence	
Total (%)	150 (26)	243 (40)		190 (34)	
Mean Score (±SD)	25 (±1.71)	29 (±1.14)		34 (±1.84)	
		β (SE)	*p*-value	β (SE)	*p*-value
Crude model	Ref	−0.040 (0.034)	0.247	−0.106 (0.036)	0.004
Model 1 ^a^	Ref	−0.050 (0.034)	0.148	−0.103 (0.036)	0.005
Model 2 ^b^	Ref	−0.046 (0.034)	0.176	−0.097 (0.037)	0.008
Model 3 ^c^	Ref	−0.051 (0.033)	0.127	−0.095 (0.035)	0.008

^a^ Adjusted for age, sex, race and education. ^b^ Adjusted for Model 1 in addition to physical activity, smoking status and energy intake. ^c^ Adjusted for Model 1 and Model 2 in addition to BMI, multimorbidity and medication use.

**Table 3 nutrients-16-01385-t003:** Linear regression results between low-grade systemic inflammation and degrees of adherence by age group. São Paulo, Brazil, 2015.

	Degrees of Adherence (MedDietScore) by Younger Adults				
	Low Adherence	Moderate Adherence		High Adherence	
Total (%)	78 (26.5)	121 (40)		91 (33.5)	
Mean score (±SD)	25 (±1.77)	30 (±1.14)		34 (±1.75)	
		β (SE)	*p*-value	β (SE)	*p*-value
Crude model	Ref	−0.005 (0.046)	0.911	−0.080 (0.049)	0.107
Model 1 ^a^	Ref	−0.012 (0.050)	0.789	−0.087 (0.050)	0.080
Model 2 ^b^	Ref	−0.010 (0.047)	0.829	−0.082 (0.050)	0.105
Model 3 ^c^	Ref	−0.010 (0.043)	0.809	−0.064 (0.046)	0.161
	**Degrees of Adherence (MedDietScore) by Older Adults**				
	**Low Adherence**	**Moderate Adherence**		**High Adherence**	
Total (%)	106 (36)	112 (38)		75 (26)	
Mean score (±SD)	26 (±1.88)	30 (±1.15)		34 (±1.82)	
		β (SE)	*p*-value	β (SE)	*p*-value
Crude model	Ref	−0.037 (0.049)	0.453	−0.139 (0.055)	0.012
Model 1 ^a^	Ref	−0.043 (0.049)	0.380	−0.118 (0.055)	0.033
Model 2 ^b^	Ref	−0.038 (0.049)	0.444	−0.119 (0.056)	0.033
Model 3 ^c^	Ref	−0.041 (0.049)	0.404	−0.115 (0.055)	0.040

^a^ Adjusted for sex, race and education. ^b^ Adjusted for Model 1 in addition to physical activity, smoking status and energy intake. ^c^ Adjusted for Model 1 and Model 2 in addition to BMI, multimorbidity and medication use.

**Table 4 nutrients-16-01385-t004:** Description of the consumption of each MedDietScore’s food component according to the degree of adherence by age group. São Paulo, Brazil, 2015.

Food Component	Food Component Intake in Each Degree of Adherence *			*p*-Value **
	Low	Moderate	High	
Non-refined cereals and potatoes				
Younger adults	3 servings/week	6 servings/week	8 servings/week	<0.001
Older adults	3 servings/week	5 servings/week	7 servings/week	<0.001
Fruits				
Younger adults	5 servings/week	8 servings/week	8 servings/week	0.021
Older adults	9 servings/week	10 servings/week	12 servings/week	0.023
Vegetables				
Younger adults	8 servings/week	10 servings/week	13 servings/week	<0.001
Older adults	9 servings/week	11 servings/week	12 servings/week	<0.001
Legumes				
Younger adults	10 servings/week	9 servings/week	8 servings/week	0.706
Older adults	9 servings/week	8 servings/week	9 servings/week	0.569
Fish				
Younger adults	1 serving/month	1 serving/month	1 serving/month	0.038
Older adults	1 serving/month	1 serving/month	1 serving/month	<0.001
Meat				
Younger adults	6 servings/week	6 servings/week	6 servings/week	0.617
Older adults	7 servings/week	6 servings/week	6 servings/week	0.044
Poultry				
Younger adults	4 servings/week	3 servings/week	2 servings/week	0.002
Older adults	3 servings/week	3 servings/week	3 servings/week	0.013
Full-fat dairy				
Younger adults	7 servings/week	6 servings/week	7 servings/week	0.374
Older adults	7 servings/week	7 servings/week	5 servings/week	0.013
Olive oil				
Younger adults	1 serving/week	1 serving/week	3 servings/week	<0.001
Older adults	1 serving/week	2 servings/week	3 servings/week	<0.001
Wine				
Younger adults	30 mL/month	50 mL/month	50 mL/month	0.006
Older adults	40 mL/month	40 mL/month	50 mL/month	0.335

* Food servings based on the 2006 Food Guide for the Brazilian population [30]. ** Kruskal–Wallis test.

**Table 5 nutrients-16-01385-t005:** Association between the systemic inflammation and cutoff MedDietScore’s food intake defined by the intake in the degrees of adherence by age groups. São Paulo, Brazil, 2015.

MedDietScore’s Food Intake	LGSI Score	
	Crude Model β (*p*)	Adjusted Model * β (*p*)
Non-refined cereals/potatoes		
Younger adults		
≤3 servings/week	Ref	Ref
>3 and <8 servings/week	−0.035 (0.447)	−0.020 (0.644)
≥8 servings/week	−0.087 (0.085)	−0.064 (0.189)
Older adults		
≤3 servings/week	Ref	Ref
>3 and <7 servings/week	−0.072 (0.153)	−0.084 (0.101)
≥7 servings/week	−0.114 (0.044)	−0.082 (0.161)
Fruits		
Younger adults		
≤5 servings/week	Ref	Ref
>5 and <8 servings/week	−0.017 (0.756)	0.010 (0.845)
≥8 servings/week	0.018 (0.661)	0.004 (0.910)
Older adults		
≤9 servings/week	Ref	Ref
>9 and <12 servings/week	−0.063 (0.268)	−0.032 (0.572)
≥12 servings/week	−0.072 (0.133)	−0.006 (0.901)
Vegetables		
Younger adults		
≤8 servings/week	Ref	Ref
>8 and <13 servings/week	−0.059 (0.200)	−0.027 (0.527)
≥13 servings/week	−0.044 (0.336)	−0.025 (0.558)
Older adults		
≤9 servings/week	Ref	Ref
>9 and <12 servings/week	−0.121 (0.029)	−0.124 (0.030)
≥12 servings/week	−0.148 (0.002)	−0.132 (0.010)
Legumes		
Younger adults		
≤8 servings/week	Ref	Ref
>8 servings/week	−0.022 (0.565)	0.023 (0.548)
Older adults		
≤9 servings/week	Ref	Ref
>9 servings/week	−0.025 (0.552)	−0.015 (0.745)
Fish		
Younger adults		
≤1 serving/month	Ref	Ref
>1 serving/month	−0.024 (0.537)	−0.001 (0.977)
Older adults		
≤1 serving/month	Ref	Ref
>1 serving/month	0.030 (0.486)	0.053 (0.228)
Red meat		
Younger adults		
≤6 servings/week	Ref	Ref
>6 servings/week	−0.008 (0.822)	0.004 (0.910)
Older adults		
≤6 servings/week	Ref	Ref
> 6 servings/week	−0.027 (0.532)	−0.059 (0.182)
Poultry		
Younger adults		
≤2 servings/week	Ref	Ref
>2 servings/week	−0.107 (0.015)	−0.065 (0.108)
Older adults		
≤3 servings/week	Ref	Ref
>3 servings/week	0.021 (0.629)	0.026 (0.556)
Full-fat dairy		
Younger adults		
≤7 servings/week	Ref	Ref
>7 servings/week	0.032 (0.389)	0.031 (0.382)
Older adults		
≤5 servings/week	Ref	Ref
>5 servings/week	0.079 (0.077)	0.082 (0.069)
Olive oil		
Younger adults		
≤1 serving/week	Ref	Ref
>1 and <3 servings/week	−0.032 (0.470)	−0.026 (0.536)
≥3 servings/week	−0.053 (0.254)	−0.034 (0.433)
Older Adults		
≤1 serving/week	Ref	Ref
>1 and <3 servings/week	−0.055 (0.267)	−0.054 (0.289)
≥3 servings/week	−0.102 (0.059)	−0.115 (0.034)
Wine		
Younger adults		
≤50 mL/month	Ref	Ref
>50 mL/month	−0.034 (0.373)	0.011 (0.756)
Older adults		
≤50 mL/month	Ref	Ref
>50 mL/month	−0.040 (0.346)	−0.042 (0.337)

* Adjusted for age, sex, race, education, physical activity, smoking status, energy intake, BMI, multimorbidity and medication use.

## Data Availability

The data presented in this study are available on request from the corresponding author. The data are not publicly available due to privacy.

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
