# Peer review of "High Adherence to the Mediterranean Dietary Pattern Is Inversely Associated with Systemic Inflammation in Older but Not in Younger Brazilian Adults"

_nutrients, 2024, doi:10.3390/nu16091385_

Round 1

Reviewer 1 Report

Comments and Suggestions for Authors

Dear authors, congratulations on this research. 

Having read it, I suggest that the following changes be implemented: 

In the theoretical framework it remains very brief. More studies need to be applied. Here are some that could be valid: 

https://doi.org/10.3390/nu12092630

https://doi.org/10.31083/j.jomh1804100 

https://doi.org/10.3390/socsci12030113 

At the end of the introduction add the research hypotheses. 

In the material and method, could you add the average age of the participants?

The discussion section should be separated from the conclusion and limitations. Add two new sections: Limitations and future perspectives and conclusions.

Adapt the references to the journal's guidelines.

Comments on the Quality of English Language

Revise the wording of the manuscript. There are some sentences that are difficult to understand. 

Reviewer 2 Report

Comments and Suggestions for Authors

The Manuscript (ID: nutrients-2746697), entitled “High adherence to Mediterranean dietary pattern is inversely associated with systemic inflammation in older but not in younger Brazilian adults”, evaluated the role of Mediterranean dietary pattern associated with systemic inflammation in Brazilian population.

Authors enrolled a large number of younger adults (n = 290) and older adults (n = 293) to assess inflammatory biomarkers, dietary consumption, socio-demographic characteristics and lifestyle variables.

The Manuscript is well written, and the quality of figures and tables are high.

Specific comments:

In the Abstract section please delete coefficients and p values.

Line 50 in the Introduction Authors should describe the use of Mediterranean aromatic herbs and spices in the salty taste (Rosa et al., Nutrients 2022, 14, 4976. https://doi.org/10.3390/nu14234976;

Rosa et al., 2022. DOI 10.1002/jsfa.11953)

Line 132 Please explain the acronym.

Please use the same definition for 24HDR or 24hDR in all text (Lines 132 and 136).

Comments on the Quality of English Language

Minor editing of English language is required

Round 2

Reviewer 1 Report

Comments and Suggestions for Authors

The paper has been improved. It can be published